# Reconstruction for Generation: Regularizing Motion Diffusion Models with Motion Reconstruction

## Abstract

Diffusion models have seen widespread adoption for text-driven human motion generation and related tasks due to their impressive generative capabilities and flexibility. However, current motion diffusion models face two major limitations: a representational gap caused by pre-trained text encoders that lack motion-specific information, and error accumulation during the iterative denoising process. This paper introduces **MO**tion **R**econstruction for **GEN**eration (**MORGEN**) to address these challenges. First, MORGEN leverages a motion latent space as intermediate supervision for text-to-motion generation. To this end, MORGEN co-trains a motion reconstruction branch with two key objective functions: self-regularization to enhance the discrimination of the motion space and motion-centric latent alignment to enable accurate mapping from text to the motion latent space. Second, we propose Reconstructive Error Guidance (REG), a testing-stage guidance mechanism that exploits the diffusion model's inherent self-correction ability to mitigate error accumulation. At each denoising step, REG uses the motion reconstruction branch to reconstruct the previous estimate, reproducing the prior error patterns. By amplifying the residual between the current prediction and the reconstructed estimate, REG highlights the improvements in the current prediction. Extensive experiments demonstrate that MORGEN achieves significant improvements and state-of-the-art performance. Our code will be released.

## 1 Introduction

Imagine giving a textual description and immediately witnessing a lifelike avatar execute it with physically plausible and faithful body movements in the correct sequence. This vision drives human motion generation with applications in virtual reality (Du et al., 2023), game content creation (Liang et al., 2024a), and embodied robotics (Xia et al., 2021). The task is inherently challenging because language is abstract while motion is continuous, high-dimensional, and kinematically constrained—demanding both fine-grained semantic understanding and robust many-to-many mappings between natural language and human motion dynamics.

This challenge has sparked extensive research interest, which can be broadly categorized into two main approaches: VQ-VAE-based and diffusion-based methods. Among these, diffusion-based methods have gained widespread adoption across downstream tasks, including motion in-betweening (Cohan et al., 2024), human-object interaction (Li et al., 2024), and human-human interaction modeling (Liang et al., 2024b), owing to their exceptional flexibility and controllability. Existing diffusion-based methods typically leverage pre-trained text encoders to obtain robust textual embeddings, such as T5 (Ni et al., 2021), CLIP (Radford et al., 2021), and DistilBERT (Sanh et al., 2019). Conditioned on these textual embeddings, motion diffusion models learn to recover motion data from noise through iterative denoising processes. Recent advances have incorporated various techniques, including latent diffusion (Chen et al., 2023), preference optimization (Sheng et al., 2024), hierarchical semantic graphs (Jin et al., 2023), and retrieval-augmented generation (Zhang et al., 2023b), which have achieved notable improvements in inference speed, motion realism, and semantic-motion alignment.

Figure 1: At inference time, MORGEN first maps a textual description onto a motion-centric latent manifold and then predicts using a diffusion model. Meanwhile, it reconstructs previous estimates that contain error patterns. By contrasting these predictions, MORGEN uses the reconstruction as a negative reference to drive the output away from poor estimates and towards the real data manifold.

Nevertheless, motion diffusion models still face severe limitations in both text models and the denoising process. First, pre-trained text models typically lack motion-specific information. While CLIP captures visual concepts that correlate with actions, it fails to encode essential temporal dynamics and kinematic constraints, having been trained exclusively on static image-text pairs. This absence forces models to bridge an unnecessarily large representational gap, hindering the learning of accurate semantic-to-dynamic mappings. Second, diffusion models suffer from error accumulation (Chung et al., 2022). More specifically, early denoising steps, which must recover motion from nearly pure noise, are particularly prone to generating error patterns. Once such artifacts emerge, they can implicitly propagate across subsequent denoising steps, leading to degraded sample quality.

Herein, we introduce **MO**tion **R**econstruction for **GEN**eration (**MORGEN**), a novel diffusion-based framework to address these challenges. For the first problem, we leverage the latent space learned through motion reconstruction as an intermediate supervision for text-to-motion generation. Specifically, MORGEN employs a two-stream pipeline (Ahuja & Morency, 2019): motion reconstruction—where the diffusion model reconstructs motion sequences conditioned on motion-encoder latents; and text-to-motion generation—where the same diffusion model generates motion from text-encoder latents. Based on this pipeline, MORGEN innovatively incorporates two objectives: (a) **self-regularization**, which computes a cross-entropy loss in the motion latent space to enhance discrimination between motion latents, helping to learn a compact yet expressive motion representation; and (b) **motion-centric latent alignment**, aligning the text latent space with the motion latent space, with carefully designed gradients to ensure stable end-to-end training. These designs together enable MORGEN to map text embeddings into a motion-aware latent space, inherently embedding the dynamic features required for realistic motion synthesis and bridging the representation gap.

To address the second problem, we introduce **R**econstructive **E**rror **G**uidance (**REG**), which harnesses the self-correcting ability of diffusion models to mitigate error accumulation. Our core insight is that diffusion models can inherently self-correct, which is similar to how they restore clean data from noise. To maximize this property, at each denoising step in the testing stage, the motion reconstruction branch reconstructs the previous estimate, capturing the earlier error patterns. We then calculate the residual between the current text-driven prediction and this reconstruction, and integrate the residual into the prediction to generate the final output. This residual highlights the improvements in the current prediction. By amplifying this term, REG directs the sampling process away from error-prone regions, thereby reducing error accumulation and enhancing the quality of generated motions throughout denoising.

By integrating these core innovations, MORGEN enables the generation of more realistic and semantically aligned motions from text. Extensive experiments show that MORGEN achieves significant improvements and state-of-the-art performance: on the HumanML3D dataset (Guo et al., 2022), MORGEN achieves an R-Precision@1 of 56.3% and an FID of 0.037 with only 20 inference steps. Consistent performance gains are also observed on the KIT-ML dataset (Plappert et al., 2016). Comprehensive ablation studies further confirm that each component makes a meaningful contribution to the overall performance improvements.

## 2 RELATED WORK

**Text-Driven Human Motion Generation.** Current research on text-to-motion generation has consolidated mainly around two principal families: diffusion models and vector-quantized variational

autoencoders (VQ-VAE). Early diffusion-based approaches such as Motion Diffusion Model (Tevet et al., 2022c) and MotionDiffuse (Zhang et al., 2022) trained denoising networks directly in the raw motion space, followed by a series of extensions that target finer semantic alignment (Zhang et al., 2023c), open-vocabulary coverage (Liang et al., 2024a), retrieval-enhanced consistency (Zhang et al., 2023b), or keyframe-centric stability (Bae et al., 2025). In parallel, latent diffusion methods first encode motions into a continuous latent space and perform denoising there, aiming for improved efficiency and quality, e.g., MLD (Chen et al., 2023), MotionLCM (Dai et al., 2024), Salad (Hong et al., 2025). Conversely, VQ-VAE-based pipelines—pioneered by T2M-GPT (Zhang et al., 2023a) and advanced through MMM (Pinyoanuntapong et al., 2024b), MoMask (Guo et al., 2024), BAMM (Pinyoanuntapong et al., 2024a), MoGenTS (Yuan et al., 2024), BAD (Hosseyni et al., 2025), KinMo (Zhang et al., 2025), and LaMP (Li et al., 2025)—have empirically exhibited higher motion fidelity, typically reflected in lower FID scores than diffusion counterparts. In this paper, MORGEN demonstrates that diffusion-based approaches can achieve FID performance comparable to that of VQ-based approaches.

**Pre-trained Text Models and Two-Stream Methods.** Since text-to-motion datasets are significantly smaller than typical text or text–image datasets, most methods utilize pre-trained text models to extract robust text embeddings. CLIP is widely used for its visual-textual embedding space (Tevet et al., 2022a), but recent research suggests it may not be optimal for aligning text and motion. Instead, these studies suggest fine-tuning text encoders to learn a joint language–motion embedding space explicitly (Maldonado et al., 2025; Zhang et al., 2025). This approach can be traced back to early two-stream methods (Ahuja & Morency, 2019), which utilize dual branches—motion reconstruction and text-to-motion generation—and share a decoder to learn a joint language-motion space implicitly. Subsequent works further constrain this space using latent alignment, KL divergence, or contrastive learning (Ghosh et al., 2021; Petrovich et al., 2022; 2023). Our method is inspired by these approaches but differs fundamentally: we center the alignment on a carefully designed motion latent space, with the text space aligning to it. We demonstrate that, when employing a diffusion model as the decoder, focusing on modeling detailed motion dynamics yields better motion synthesis results than forcing the learning of a joint language-motion space.

**Diffusion Guidance.** Guidance in diffusion sampling typically combines multiple score estimates to enrich the effective target distribution or to impose auxiliary conditioning (Dhariwal & Nichol, 2021; Ho & Salimans, 2022; Karras et al., 2024). Common estimates include conditional score estimates $\nabla_{x_t} \log p(x_t|t, c)$, unconditional score estimates $\nabla_{x_t} \log p(x_t|t)$, classifier gradients $\nabla_{x_t} \log p(y|x_t)$, and CLIP-derived similarity gradients (Dhariwal & Nichol, 2021; Nichol et al., 2021; Ho & Salimans, 2022). Recent works further introduce deliberately weakened auxiliary scores by degrading the predictor—e.g., applying dropout (Karras et al., 2024), skipping layers (Stability AI, 2024), or perturbing attention (Ahn et al., 2024). These weak scores function as contrastive references: amplifying samples favored by stronger scores while suppressing those aligned with weaker ones improves fidelity and semantic alignment. In the same spirit, we derive a weakened motion-conditioned score by conditioning the predictor on a motion latent that carries previously introduced error patterns, and use it as a contrastive reference within our guidance mechanism.

## 3 METHOD

**Overview.** MORGEN generates a sequence of realistic human motion from a given text description. This process starts by extracting text embeddings with pre-trained text models. These embeddings are mapped onto a motion latent manifold and decoded into a motion sequence using a diffusion model. MORGEN also reconstructs the past motion estimate as a negative reference in the inference stage, as illustrated in Figure 1. By guiding predictions away from this reference, MORGEN achieves improved sampling quality.

For a thorough understanding of MORGEN, we begin by presenting the overall architecture, which features two main branches: motion reconstruction and text-to-motion generation (Section 3.1). Next, we detail the training objectives, introducing self-regularization and motion-centric latent alignment, which facilitate learning an expressive motion latent space and enable effective mapping from text to motion latents (Section 3.1). Lastly, we provide an in-depth explanation of Reconstructive Error Guidance and inference sampling (Section 3.3). Figure 2 provides an overview.

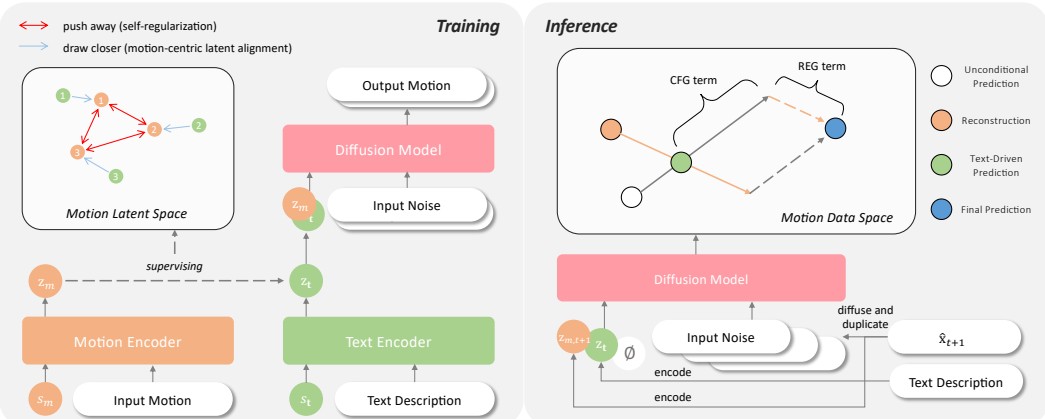

Figure 2: **Overview of MORGEN.** During training, MORGEN learns a motion latent space through motion reconstruction, with self-regularization to encourage better separability between motion latents, resulting in improved semantic resolution. The text latents from the text encoder are drawn closer to corresponding motion latents through motion-centric latent alignment. At each inference step, given the last step prediction $\hat{\mathbf{x}}_{t+1,s}$ and text description, MORGEN first encodes them into latents $\mathbf{z}_{m,t+1}$ and $\mathbf{z_t}$. Then, these latents, together with a zero vector $\varnothing$ and input noise, are fed into diffusion motion to obtain reconstruction, text-driven prediction, and unconditional prediction. These outputs are combined to produce the final output.

## 3.1 MORGEN ARCHITECTURE

Given a motion sequence $\mathbf{x}_0 \in \mathbb{R}^{T \times d}$ or a text description $\mathbf{t}$, MORGEN either reconstructs the input motion or generates a motion sequence to match a given description. This is achieved through two branches: motion reconstruction and text-to-motion generation, both of which share the Motion Diffusion Model (MDM) (Shafir et al., 2023) as the decoder.

**Motion Diffusion Model.** MDM is modeled as a Markov noising chain $\{\mathbf{x}_t\}_{t=0}^T$ with $\mathbf{x}_0$ drawn from the data distribution. The forward diffusion process incrementally adds Gaussian noise: $q(\mathbf{x}_t|\mathbf{x}_{t-1}) = \mathcal{N}\big(\mathbf{x}_t; \sqrt{1-\beta_t}\,\mathbf{x}_{t-1}, \beta_t I\big)$. In the reverse process, a denoiser $D$ learns to recover clean motion from a noisy input $\mathbf{x}_t$: $\hat{\mathbf{x}}_t = D(\mathbf{x}_t, t, c)$, where $\hat{\mathbf{x}}_t$ is the motion estimate at timestep $t$ and $c$ denotes the conditioning.

**Motion Reconstruction.** The motion reconstruction branch encodes $\mathbf{x}$ into a motion latent $\mathbf{z}_m$ using a transformer-based motion encoder $E_m(\cdot)$, which takes a special token $\mathbf{s}_m$ and the motion sequence as input. The output $\mathbf{z}_m$ represents the global concept of the sequence. The diffusion decoder $D$ takes $\mathbf{z}_m$, timestep $t$, and noisy motion $\mathbf{x}_t$ to predict the clean motion. This process can be expressed as:

$$\mathbf{z}_m = E_m(\mathbf{s}_m, \mathbf{x}_0), \quad \hat{\mathbf{x}}_0 = D(\mathbf{x}_t, t, \mathbf{z}_m). \tag{1}$$

Here $t$ denotes the diffusion timestep, which is sampled uniformly as $t \sim \mathcal{U}\{0, \ldots, T-1\}$, where $T$ is the total number of diffusion steps.

**Text-to-Motion Generation.** The text-to-motion branch mirrors the motion reconstruction branch, encoding the text embedding $\mathbf{f_t}$ with a text encoder $E_\mathbf{t}$ and then decoding with $D$:

$$\mathbf{z_t} = E_\mathbf{t}(\mathbf{s_t}, \mathbf{f_t}), \quad \hat{\mathbf{x}}_0 = D(\mathbf{x}_t, t, \mathbf{z_t}). \tag{2}$$

Here, $\mathbf{f_t} \in \mathbb{R}^{L \times d_f}$ is the token-level text embedding extracted from $\mathbf{t}$ (where $L$ is sequence length), $\mathbf{s}_t$ is the special input token, and $\mathbf{z_t}$ is the latent produced by the text encoder $E_\mathbf{t}$.

## 3.2 OPTIMIZATION OBJECTIVES

The training objectives of MORGEN consist of four key components: reconstruction, text-driven generation, self-regularization, and motion-centric latent alignment. For clarity, we divide these objectives into two categories: (1) reconstruction and text-driven generation, which follow established

two-stream approaches (Ahuja & Morency, 2019; Petrovich et al., 2022), and (2) self-regularization and motion-centric latent alignment, which are our novel contributions aimed at learning a compact yet expressive motion latent space and enabling the effective mapping from text to motion latents. Below, we provide a detailed introduction to the formulation and specific function of each objective.

**Reconstruction.** This objective encourages the diffusion model, given the motion latent $\mathbf{z}_m$, timestep $t$, and noisy motion $\mathbf{x}_t$, to accurately reconstruct the input motion sequence:

$$L_{\mathrm{rec}} = \mathbb{E}_{\mathbf{x}_0,t}\big[\|D(\mathbf{x}_t, t, \mathbf{z}_m) - \mathbf{x}_0\|_2^2\big] = \mathbb{E}_{\mathbf{x}_0,t}\big[\|D(\mathbf{x}_t, t, E_m(\mathbf{s}_m, \mathbf{x}_0)) - \mathbf{x}_0\|_2^2\big]. \tag{3}$$

This loss jointly trains both the diffusion model and the motion encoder, aiming for a strong motion decoder, a motion encoder that extracts abstract representations of motion, and a compact latent space with essential motion dynamics.

**Text-Driven Generation.** In this objective, the diffusion model learns to generate motion conditioned on the text latent $\mathbf{z_t}$, timestep $t$, and noisy motion $\mathbf{x}_t$:

$$L_{\mathrm{gen}} = \mathbb{E}_{\mathbf{x}_0,t,\mathbf{t}}\big[\|D(\mathbf{x}_t, t, \mathbf{z_t}) - \mathbf{x}_0\|_2^2\big] = \mathbb{E}_{\mathbf{x}_0,t}\big[\|D(\mathbf{x}_t, t, E_\mathbf{t}(\mathbf{s_t}, \mathbf{f_t})) - \mathbf{x}_0\|_2^2\big]. \tag{4}$$

This objective encourages the diffusion model to adapt to conditioning on the text latent manifold, since there are inherent differences between the text and motion manifolds.

**Self-Regularization.** This objective can be viewed as a cross-entropy loss operating on the motion latent space. For a batch of size $B$, let the normalized motion latents be $\tilde{\mathbf{z}}_m^i$, and define the similarity $\mathrm{sim}(\tilde{\mathbf{z}}_m^i, \tilde{\mathbf{z}}_m^j) = (\tilde{\mathbf{z}}_m^i)^\top \tilde{\mathbf{z}}_m^j$, which corresponds to cosine similarity after normalization. With a temperature parameter $\tau = 1$, and treating only identical indices as positive pairs, the loss is defined as:

$$L_{\mathrm{sr}} = \frac{1}{B} \sum_{i=1}^{B} -\log \frac{\exp(\mathrm{sim}(\tilde{\mathbf{z}}_m^i, \tilde{\mathbf{z}}_m^i)/\tau)}{\sum_{j=1}^{B} \exp(\mathrm{sim}(\tilde{\mathbf{z}}_m^i, \tilde{\mathbf{z}}_m^j)/\tau)}. \tag{5}$$

This loss encourages better separability among motion latents, producing a broader and more expressive manifold with improved semantic resolution. Consequently, the refined latent space enables more precise mapping from text representations to motion latents in the subsequent alignment objective.

**Motion-Centric Latent Alignment.** This objective aligns the text manifold with the motion manifold. Given a paired text description and motion sequence, this objective minimizes the distance between the corresponding text latent $\mathbf{z_t}$ and motion latent $\mathbf{z}_m$:

$$L_{\mathrm{latent}} = \mathbb{E}_{\mathbf{z}_m,\mathbf{z_t}}\big[\|\mathbf{z_t} - (1 - \beta)\,\mathrm{sg}(\mathbf{z}_m) - \beta\mathbf{z}_m\|_2^2\big], \tag{6}$$

where $\mathrm{sg}(\cdot)$ is the stop-gradient operator and $\beta$ modulates the flow of gradients to the motion encoder $E_m$. We set $\beta = 0.01$ so that MORGEN's latent space remains motion-centric, yet flexible enough to adapt minimally to the text space. This is based on two insights: (1) prioritizing motion space leads to stronger performance than enforcing a fully joint language-motion space, as mapping motion to text sacrifices important motion dynamics, and (2) with end-to-end training, motion latents evolve during alignment. A purely text-to-motion alignment ($\beta = 0$) makes optimization harder. Thus, a small $\beta$ supports convergence while retaining motion information.

**Overall Objective.** The final training objective is a weighted sum:

$$L_{\mathrm{overall}} = L_{\mathrm{rec}} + L_{\mathrm{gen}} + w_{\mathrm{sr}} L_{\mathrm{sr}} + w_{\mathrm{latent}} L_{\mathrm{latent}}, \tag{7}$$

where $w_{\mathrm{sr}}$ and $w_{\mathrm{latent}}$ are hyperparameters that determine the significance of the $L_{\mathrm{sr}}$ and $L_{\mathrm{latent}}$ terms, respectively. We empirically set both $w_{\mathrm{sr}}$ and $w_{\mathrm{latent}}$ as 1.

### 3.3 INFERENCE

**Reconstructive Error Guidance.** During training, diffusion models operate exclusively on the canonical data manifold, where the noised input follows $x_t = \sqrt{\bar{\alpha}_t}x_0 + \sqrt{1 - \bar{\alpha}_t}\epsilon$. However, during inference, their predictions often exhibit error patterns and drift away from this manifold. Denoising

based on such off-manifold predictions further exacerbates the deviation. Overall, diffusion models can cause the sampling path to deviate from the data manifold, resulting in degraded sampling quality (Chung et al., 2022).

We hypothesize that diffusion models possess an inherent capacity to self-correct such error patterns—a capability analogous to their fundamental ability to recover clean data from noise. However, this corrective potential requires explicit activation and guidance. To harness this intrinsic error-correction capability, we propose an intuitive approach that operates at each denoising step.

Specifically, at inference step $t$, we first reconstruct the prediction from the previous step $t + 1$ to explicitly capture the embedded error patterns. We then amplify the improvement achieved by the current step's prediction through a residual amplification mechanism. Let $\hat{\mathbf{x}}_{t+1,s}$ denote the final output at step $t + 1$. Our method can be formulated as:

$$\hat{\mathbf{x}}_{t,s} = D(\mathbf{x}_t, t, \mathbf{z_t}) + w \left( D(\mathbf{x}_t, t, \mathbf{z_t}) - D(\mathbf{x}_t, t, \mathbf{z}_{m,t+1}) \right), \tag{8}$$

where $\mathbf{z}_{m,t+1} = E_m(\mathbf{s}_m, \hat{\mathbf{x}}_{t+1,s})$ represents the reconstructed motion latent from the previous step, and $w \geq 0$ is a weighting coefficient that controls the amplification strength of the residual correction term. We term this inference strategy Reconstructive Error Guidance (REG).

**Inference Sampling.** Finally, during inference, we combine REG with the commonly used classifier-free guidance (CFG) for sampling. The final output for each denoising step $t$, denoted as $\hat{\mathbf{x}}_{t,s}$, is computed as:

$$\hat{\mathbf{x}}_{t,s} = D(\mathbf{x}_t, t, \mathbf{z_t}) + w_1 \underbrace{\left( D(\mathbf{x}_t, t, \mathbf{z_t}) - D(\mathbf{x}_t, t, \mathbf{z}_{m,t+1}) \right)}_{\text{REG term}} + w_2 \underbrace{\left( D(\mathbf{x}_t, t, \mathbf{z_t}) - D(\mathbf{x}_t, t, \varnothing) \right)}_{\text{CFG term}}, \tag{9}$$

where the final term represents the standard CFG residual between conditional and unconditional predictions (with unconditional input denoted by $\varnothing$). Here, $w_1$ and $w_2$ respectively control the influence of REG and CFG.

## 4 EXPERIMENT

### 4.1 DATASETS AND METRICS

**Datasets.** HumanML3D (Guo et al., 2022) is a large-scale text–motion dataset containing 14,616 motion sequences from AMASS (Mahmood et al., 2019), each annotated with 44,970 sequence-level textual descriptions. By comparison, the KIT dataset (Plappert et al., 2016) is smaller, offering 3,911 motion sequences and 6,353 textual descriptions. For both datasets, we use the standard redundant motion representation, which includes joint velocities, positions, and rotations.

**Metrics.** We assess the generated motions with five complementary metrics. R-Precision and Multimodal-Dist measure the semantic alignment between generated motions and text descriptions. Fréchet Inception Distance (FID) evaluates the distributional similarity between generated motions and the ground truth in a learned latent space. Diversity quantifies the variability within the generated motion set, while MultiModality Distance (MM Dist) captures the average variance among motions conditioned on the same description.

### 4.2 IMPLEMENTATION DETAILS

We adopt exactly the same text and motion encoders as those in TEMOS (Petrovich et al., 2022). Both of them are implemented as 6-layer, encoder-only transformers. The text encoder takes the text embeddings extracted by DistilBERT (Sanh et al., 2019) as input. The latent dimensionality is set to 256 for HumanML3D and 192 for KIT-ML. For the diffusion model that generates motion sequences from the latents, we use the MDM architecture (Shafir et al., 2023), consisting of an 8-layer, encoder-only Transformer backbone with a latent size of 512. Training is performed with a batch size of 64, a learning rate of 0.0001, and the AdamW optimizer. Models are trained for 450K steps on HumanML3D and 400K steps on KIT-ML. The diffusion process runs over $T = 50$ steps during training, with 10% of conditional latents replaced by zero vectors for classifier-free guidance. During inference, 20 denoising steps, spaced linearly from $[0, \ldots, T - 1]$, are used, resulting in 20 inference steps. Reconstructive Error Guidance and classifier-free guidance use weights $w_1 = 5.0$ and $w_2 = 1.5$, respectively.

### 4.3 Comparison with State-of-the-Art Methods

Table 1: Quantitative results of text-to-motion generation on the HumanML3D test set.

| | Method | FID↓ | R-Precision Top 1 | R-Precision Top 2 | R-Precision Top 3 | MM Dist↓ | Diversity↑ | MM↑ |
|---|---|---|---|---|---|---|---|---|
| | Ground Truth | $0.002^{\pm.000}$ | $0.511^{\pm.003}$ | $0.703^{\pm.003}$ | $0.797^{\pm.002}$ | $2.974^{\pm.008}$ | $9.503^{\pm.065}$ | - |
| VQ-VAE-based | T2M-GPT (Zhang et al., 2023a) | $0.116^{\pm.004}$ | $0.491^{\pm.003}$ | $0.680^{\pm.003}$ | $0.775^{\pm.002}$ | $3.118^{\pm.011}$ | $9.761^{\pm.081}$ | $1.856^{\pm.011}$ |
| | MMM (Pinyoanuntapong et al., 2024b) | $0.080^{\pm.003}$ | $0.504^{\pm.003}$ | $0.696^{\pm.003}$ | $0.794^{\pm.002}$ | $2.998^{\pm.007}$ | $9.411^{\pm.058}$ | $1.164^{\pm.041}$ |
| | MoMask (Guo et al., 2024) | $0.045^{\pm.002}$ | $0.521^{\pm.002}$ | $0.713^{\pm.002}$ | $0.807^{\pm.002}$ | $2.958^{\pm.008}$ | - | $1.241^{\pm.040}$ |
| | BAMM (Pinyoanuntapong et al., 2024a) | $0.055^{\pm.002}$ | $0.525^{\pm.002}$ | $0.720^{\pm.003}$ | $0.814^{\pm.003}$ | $2.919^{\pm.008}$ | $9.717^{\pm.089}$ | $1.687^{\pm.051}$ |
| | MoGenTS (Yuan et al., 2024) | $0.033^{\pm.001}$ | $0.529^{\pm.003}$ | $0.719^{\pm.002}$ | $0.812^{\pm.002}$ | $2.867^{\pm.006}$ | $9.570^{\pm.077}$ | - |
| | BAD (Hosseyni et al., 2025) | $0.065^{\pm.003}$ | $0.517^{\pm.002}$ | $0.713^{\pm.003}$ | $0.808^{\pm.003}$ | $2.901^{\pm.008}$ | $9.694^{\pm.068}$ | $1.194^{\pm.044}$ |
| | KinMo (Zhang et al., 2025) | $0.039^{\pm.003}$ | $0.532^{\pm.002}$ | $0.724^{\pm.003}$ | $0.821^{\pm.003}$ | $2.901^{\pm.010}$ | $9.674^{\pm.058}$ | $1.321^{\pm.039}$ |
| | LaMP (Li et al., 2025) | $0.032^{\pm.002}$ | $0.557^{\pm.003}$ | $0.751^{\pm.002}$ | $0.843^{\pm.001}$ | $2.759^{\pm.007}$ | $9.571^{\pm.069}$ | - |
| Diffusion-based | MDM (Tevet et al., 2022c) | $0.489^{\pm.025}$ | $0.418^{\pm.005}$ | $0.604^{\pm.001}$ | $0.707^{\pm.004}$ | $3.360^{\pm.023}$ | $9.450^{\pm.066}$ | $2.860^{\pm1.11}$ |
| | MLD (Chen et al., 2023) | $0.473^{\pm.013}$ | $0.481^{\pm.003}$ | $0.673^{\pm.003}$ | $0.772^{\pm.002}$ | $3.196^{\pm.010}$ | $9.724^{\pm.082}$ | $2.413^{\pm.079}$ |
| | ReMoDiffuse (Zhang et al., 2023b) | $0.103^{\pm.004}$ | $0.510^{\pm.005}$ | $0.698^{\pm.006}$ | $0.795^{\pm.004}$ | $2.974^{\pm.016}$ | $9.018^{\pm.075}$ | $1.795^{\pm.043}$ |
| | FineMoGen (Zhang et al., 2023c) | $0.151^{\pm.008}$ | $0.504^{\pm.002}$ | $0.690^{\pm.002}$ | $0.784^{\pm.002}$ | $2.998^{\pm.008}$ | $9.263^{\pm.094}$ | $2.696^{\pm.079}$ |
| | MotionLCM (Dai et al., 2024) | $0.304^{\pm.012}$ | $0.502^{\pm.003}$ | $0.698^{\pm.002}$ | $0.798^{\pm.002}$ | $3.012^{\pm.007}$ | $9.607^{\pm.066}$ | $2.259^{\pm.092}$ |
| | StableMoFusion (Huang et al., 2024) | $0.098^{\pm.003}$ | $0.553^{\pm.003}$ | $0.748^{\pm.002}$ | $0.841^{\pm.002}$ | - | $9.748^{\pm.092}$ | $1.774^{\pm.051}$ |
| | CLoSD (Tevet et al., 2022b) | $0.283^{\pm.000}$ | $0.464^{\pm.000}$ | $0.668^{\pm.000}$ | $0.777^{\pm.000}$ | $3.150^{\pm.000}$ | $9.210^{\pm.000}$ | - |
| | Salad (Hong et al., 2025) | $0.076^{\pm.002}$ | $0.581^{\pm.003}$ | $0.769^{\pm.003}$ | $0.857^{\pm.002}$ | $2.649^{\pm.009}$ | $9.696^{\pm.096}$ | $1.751^{\pm.062}$ |
| | sMDM (Bae et al., 2025) | $0.130^{\pm.000}$ | $0.494^{\pm.000}$ | $0.682^{\pm.000}$ | $0.776^{\pm.000}$ | $3.051^{\pm.000}$ | $9.663^{\pm.000}$ | - |
| | **MORGEN** (Ours, $w_{\text{latent}} = 1.0$) | $0.037^{\pm.002}$ | $0.563^{\pm.003}$ | $0.755^{\pm.002}$ | $0.843^{\pm.002}$ | $2.693^{\pm.008}$ | $9.496^{\pm.094}$ | $1.066^{\pm.038}$ |
| | **MORGEN** (Ours, $w_{\text{latent}} = 0.5$) | $0.032^{\pm.002}$ | $0.561^{\pm.003}$ | $0.751^{\pm.002}$ | $0.839^{\pm.002}$ | $2.716^{\pm.007}$ | $9.487^{\pm.084}$ | $1.142^{\pm.038}$ |

Table 2: Quantitative results of text-to-motion generation on the KIT test set.

| | Method | FID↓ | R-Precision Top 1 | R-Precision Top 2 | R-Precision Top 3 | MM Dist↓ | Diversity↑ | MM↑ |
|---|---|---|---|---|---|---|---|---|
| | Ground Truth | $0.031^{\pm.004}$ | $0.424^{\pm.005}$ | $0.649^{\pm.006}$ | $0.779^{\pm.006}$ | $2.788^{\pm.012}$ | $11.08^{\pm.097}$ | - |
| VQ-VAE-based | T2M-GPT (Zhang et al., 2023a) | $0.512^{\pm.029}$ | $0.416^{\pm.006}$ | $0.627^{\pm.006}$ | $0.745^{\pm.006}$ | $3.007^{\pm.023}$ | $10.92^{\pm.108}$ | $1.856^{\pm.011}$ |
| | MMM (Pinyoanuntapong et al., 2024b) | $0.316^{\pm.028}$ | $0.404^{\pm.005}$ | $0.621^{\pm.005}$ | $0.744^{\pm.004}$ | $2.977^{\pm.019}$ | $10.91^{\pm.101}$ | $1.232^{\pm.039}$ |
| | MoMask (Guo et al., 2024) | $0.204^{\pm.011}$ | $0.433^{\pm.007}$ | $0.656^{\pm.005}$ | $0.781^{\pm.005}$ | $2.779^{\pm.022}$ | - | $1.131^{\pm.043}$ |
| | BAMM (Pinyoanuntapong et al., 2024a) | $0.183^{\pm.013}$ | $0.438^{\pm.009}$ | $0.661^{\pm.009}$ | $0.788^{\pm.005}$ | $2.723^{\pm.026}$ | $11.01^{\pm.094}$ | $1.609^{\pm.065}$ |
| | MoGenTS (Yuan et al., 2024) | $0.143^{\pm.004}$ | $0.445^{\pm.006}$ | $0.671^{\pm.006}$ | $0.797^{\pm.005}$ | $2.711^{\pm.024}$ | $10.92^{\pm.090}$ | - |
| | BAD (Hosseyni et al., 2025) | $0.221^{\pm.012}$ | $0.417^{\pm.006}$ | $0.631^{\pm.006}$ | $0.750^{\pm.006}$ | $2.941^{\pm.025}$ | $11.00^{\pm.100}$ | $1.170^{\pm.047}$ |
| | LaMP (Li et al., 2025) | $0.141^{\pm.013}$ | $0.479^{\pm.006}$ | $0.691^{\pm.005}$ | $0.826^{\pm.005}$ | $2.704^{\pm.018}$ | $10.93^{\pm.101}$ | - |
| Diffusion-based | MDM (Tevet et al., 2022c) | $0.547^{\pm.070}$ | $0.404^{\pm.002}$ | $0.616^{\pm.013}$ | $0.737^{\pm.005}$ | $3.074^{\pm.018}$ | $10.75^{\pm.203}$ | $1.806^{\pm.180}$ |
| | MLD (Chen et al., 2023) | $0.404^{\pm.027}$ | $0.390^{\pm.008}$ | $0.609^{\pm.008}$ | $0.734^{\pm.007}$ | $3.204^{\pm.027}$ | $10.80^{\pm.117}$ | $2.192^{\pm.071}$ |
| | ReMoDiffuse (Zhang et al., 2023b) | $0.155^{\pm.006}$ | $0.427^{\pm.014}$ | $0.641^{\pm.004}$ | $0.765^{\pm.055}$ | $2.814^{\pm.012}$ | $10.80^{\pm.105}$ | $1.239^{\pm.028}$ |
| | FineMoGen (Zhang et al., 2023c) | $0.178^{\pm.007}$ | $0.432^{\pm.006}$ | $0.649^{\pm.005}$ | $0.772^{\pm.006}$ | $2.869^{\pm.014}$ | $10.85^{\pm.115}$ | $1.877^{\pm.093}$ |
| | StableMoFusion (Huang et al., 2024) | $0.258^{\pm.029}$ | $0.445^{\pm.006}$ | $0.660^{\pm.005}$ | $0.782^{\pm.004}$ | - | $10.94^{\pm.077}$ | $1.362^{\pm.062}$ |
| | Salad (Hong et al., 2025) | $0.296^{\pm.012}$ | $0.477^{\pm.006}$ | $0.711^{\pm.005}$ | $0.828^{\pm.005}$ | $2.585^{\pm.016}$ | $11.10^{\pm.095}$ | $1.004^{\pm.040}$ |
| | **MORGEN** (Ours) | $0.189^{\pm.014}$ | $0.466^{\pm.005}$ | $0.688^{\pm.006}$ | $0.801^{\pm.005}$ | $2.675^{\pm.020}$ | $11.12^{\pm.089}$ | $1.159^{\pm.051}$ |

We quantitatively compare MORGEN with state-of-the-art (SOTA) methods on HumanML3D and KIT-ML. The results are shown in Table 1 and Table 2, respectively. As demonstrated in Table 1, MORGEN achieves SOTA performance on the most widely used HumanML3D benchmark. Compared with diffusion-based methods, MORGEN shows a substantial improvement in FID and achieves near-SOTA performance in semantic accuracy as measured by R-Precision, ranking just behind Salad. When compared to VQ-VAE-based approaches, MORGEN surpasses them in semantic accuracy and achieves highly competitive FID—a feat not previously attained by diffusion-based methods. MORGEN thus demonstrates that diffusion-based motion generation models can reach state-of-the-art FID levels. On the KIT-ML dataset, whose smaller scale poses significant challenges for training motion generation models—particularly diffusion-based ones—MORGEN, like the previous best diffusion-based method Salad (Hong et al., 2025), experiences a performance drop. Nevertheless, MORGEN's results remain highly competitive within this context.

Importantly, by adjusting the latent alignment weight $w_{\text{latent}}$, MORGEN can reach either state-of-the-art FID or achieve even better semantic accuracy (in terms of R-Precision and MM Dist). For our final model, we use $w_{\text{latent}} = 1.0$ as a balanced choice. Additional experiments on the impact of weight selection are detailed in Appendix A.3.

### 4.4 Ablation Studies

To assess the impact of key design choices within MORGEN, we conduct comprehensive ablation studies on HumanML3D. Specifically, these study includes: (1) *Incremental Experiments*—starting from a baseline model, we progressively introduce key design components, culminating in the complete MORGEN; (2) *Loss Hyperparameter Analysis*—we investigate the effects of loss function hyperparameters $\beta$ and $\tau$ for latent alignment and self-regularization; and (3) *Guidance Evaluation*—we examine the effectiveness of our proposed Reconstructive Error Guidance and the additional benefits achieved when it is combined with classifier-free guidance (CFG).

Table 3: Incremental experiments on key designs within MORGEN.

| Components | | | | FID↓ | R-Precision | | | MM Dist↓ | Diversity↑ |
| $E_m$ | $L_{\text{latent}}$ | $L_{\text{sr}}$ | REG | | Top 1 | Top 2 | Top 3 | | |
|---|---|---|---|---|---|---|---|---|---|
| | | | | $0.786^{\pm.016}$ | $0.417^{\pm.002}$ | $0.613^{\pm.002}$ | $0.729^{\pm.003}$ | $3.433^{\pm.012}$ | $10.063^{\pm.076}$ |
| ✓ | | | | $0.624^{\pm.013}$ | $0.493^{\pm.004}$ | $0.695^{\pm.002}$ | $0.800^{\pm.002}$ | $3.045^{\pm.013}$ | $10.188^{\pm.096}$ |
| ✓ | ✓ | | | $0.243^{\pm.009}$ | $0.527^{\pm.003}$ | $0.719^{\pm.002}$ | $0.812^{\pm.002}$ | $2.896^{\pm.008}$ | $9.703^{\pm.086}$ |
| ✓ | ✓ | ✓ | | $0.126^{\pm.005}$ | $0.560^{\pm.003}$ | $0.751^{\pm.002}$ | $0.842^{\pm.002}$ | $2.720^{\pm.007}$ | $9.689^{\pm.095}$ |
| ✓ | ✓ | ✓ | ✓ | $0.037^{\pm.002}$ | $0.563^{\pm.003}$ | $0.755^{\pm.002}$ | $0.843^{\pm.002}$ | $2.693^{\pm.008}$ | $9.496^{\pm.094}$ |

Table 4: Effect of hyperparameters $\beta$ and $\tau$.

| $\beta$ | $\tau$ | FID↓ | R-Precision | | | MM Dist↓ | Diversity↑ |
| | | | Top 1 | Top 2 | Top 3 | | |
|---|---|---|---|---|---|---|---|
| 1.00 | 1.0 | $0.336^{\pm.005}$ | $0.492^{\pm.003}$ | $0.693^{\pm.003}$ | $0.796^{\pm.002}$ | $3.047^{\pm.010}$ | $9.849^{\pm.080}$ |
| 0.10 | 1.0 | $0.136^{\pm.005}$ | $0.534^{\pm.002}$ | $0.732^{\pm.002}$ | $0.826^{\pm.002}$ | $2.835^{\pm.009}$ | $9.742^{\pm.088}$ |
| 0.01 | 1.0 | $0.037^{\pm.002}$ | $0.563^{\pm.003}$ | $0.755^{\pm.002}$ | $0.843^{\pm.002}$ | $2.693^{\pm.008}$ | $9.496^{\pm.094}$ |
| 0.00 | 1.0 | $0.051^{\pm.003}$ | $0.557^{\pm.002}$ | $0.747^{\pm.003}$ | $0.837^{\pm.002}$ | $2.712^{\pm.008}$ | $9.424^{\pm.079}$ |
| 0.01 | 2.0 | $0.044^{\pm.002}$ | $0.562^{\pm.003}$ | $0.755^{\pm.002}$ | $0.841^{\pm.002}$ | $2.701^{\pm.007}$ | $9.575^{\pm.081}$ |
| 0.01 | 1.0 | $0.037^{\pm.002}$ | $0.563^{\pm.003}$ | $0.755^{\pm.002}$ | $0.843^{\pm.002}$ | $2.693^{\pm.008}$ | $9.496^{\pm.094}$ |
| 0.01 | 0.5 | $0.044^{\pm.002}$ | $0.556^{\pm.002}$ | $0.748^{\pm.002}$ | $0.838^{\pm.002}$ | $2.726^{\pm.007}$ | $9.484^{\pm.081}$ |

**Incremental Experiments.** Table 3 presents the results of our incremental ablation studies. To rigorously assess the contribution of each component, we begin with a clean baseline consisting of MORGEN's text encoder and diffusion model only. We keep these modules exactly the same as those in MORGEN and progressively add key components. The baseline demonstrates limited performance, partly due to the challenging inference setting of only 20 steps. Introducing the motion encoder $E_m$—which forms a dual-branch architecture, similar to a direct application of Ahuja & Morency (2019) to diffusion models—provides only a modest improvement, suggesting that implicitly learning a joint language-motion space is of limited effectiveness. Incorporating $L_{\text{latent}}$ delivers substantial further gains, though still falls short of state-of-the-art performance. Adding $L_{\text{sr}}$ leads to results that surpass most diffusion-based approaches reported in Table 1. Finally, enabling REG elevates MORGEN to state-of-the-art performance. Collectively, these findings demonstrate the necessity and effectiveness of each design choice.

**Hyperparameter Analysis.** Table 4 presents the results of our hyperparameter analysis. In this table, $\beta$ controls the gradient flow from the latent alignment loss $L_{\text{latent}}$ to the motion encoder $E_m$, with $\beta = 0$ fully blocking the gradient and $\beta = 1$ allowing unrestricted gradient flow. Our results show that allowing equal proximity between text and motion latents ($\beta = 1$) is suboptimal, as this alignment comes at the expense of motion information in the motion latents. In fact, reducing the gradient flow to $E_m$ improves performance, with the best results achieved at $\beta = 0.01$. We attribute this to the constant evolution of the motion latent space during training, which increases the difficulty of latent alignment. By setting $\beta = 0.01$, we ease the alignment process while preserving essential motion information in the latent space. Another parameter shown in the Table 4, $\tau$, determines the sharpness of similarity in $L_{\text{sr}}$. A smaller $\tau$ produces sharper similarities, pushing motion latents farther apart; if too extreme, this can distort the structure of the motion manifold. In contrast, a larger $\tau$ smooths the similarity, relaxing the constraints between latents, but may reduce gains in semantic resolution. Empirically, we found $\tau = 1$ offers a desirable balance.

**Guidance Evaluation.** Table 5 presents the results of our guidance evaluation. The findings show that classifier-free guidance (CFG) substantially improves semantic accuracy, as measured by R-Precision. In contrast, our proposed reconstructive error guidance (REG) notably enhances the overall realism of the generated motion, reflected by lower FID scores. Furthermore, combining both strategies enables MORGEN to achieve state-of-the-art performance.

### 4.5 QUALITATIVE ANALYSIS

We compare the qualitative results of MORGEN with those generated by MDM (Shafir et al., 2023), MoMask (Guo et al., 2024), and Salad (Hong et al., 2025). Figure 3 illustrates three groups of comparisons, with the input text descriptions shown below each group. As each text prompt consists of multiple actions, this setup poses a significant challenge for accurate motion generation. It can be

Table 5: Effect of Reconstructive Error Guidance (REG) and classifier-free guidance (CFG). $w_1$ and $w_2$ respectively control the influence of REG and CFG.

| $w_1$ | $w_2$ | FID↓ | R-Precision | | | MM Dist↓ | Diversity↑ |
| | | | Top 1 | Top 2 | Top 3 | | |
| --- | --- | --- | --- | --- | --- | --- | --- |
| 0.0 | 0.0 | $0.293^{\pm.009}$ | $0.531^{\pm.003}$ | $0.723^{\pm.002}$ | $0.817^{\pm.002}$ | $2.875^{\pm.008}$ | $9.690^{\pm.092}$ |
| 3.0 | 0.0 | $0.075^{\pm.003}$ | $0.548^{\pm.003}$ | $0.741^{\pm.002}$ | $0.831^{\pm.002}$ | $2.766^{\pm.008}$ | $9.431^{\pm.084}$ |
| 4.0 | 0.0 | $0.072^{\pm.003}$ | $0.545^{\pm.002}$ | $0.740^{\pm.002}$ | $0.829^{\pm.002}$ | $2.772^{\pm.007}$ | $9.361^{\pm.082}$ |
| 5.0 | 0.0 | $0.088^{\pm.003}$ | $0.541^{\pm.003}$ | $0.737^{\pm.003}$ | $0.826^{\pm.002}$ | $2.791^{\pm.007}$ | $9.300^{\pm.085}$ |
| 0.0 | 1.5 | $0.126^{\pm.005}$ | $0.560^{\pm.003}$ | $0.751^{\pm.002}$ | $0.842^{\pm.002}$ | $2.720^{\pm.007}$ | $9.689^{\pm.095}$ |
| 0.0 | 2.5 | $0.106^{\pm.005}$ | $0.560^{\pm.003}$ | $0.753^{\pm.002}$ | $0.843^{\pm.002}$ | $2.710^{\pm.006}$ | $9.639^{\pm.096}$ |
| 0.0 | 3.5 | $0.101^{\pm.004}$ | $0.560^{\pm.003}$ | $0.753^{\pm.002}$ | $0.842^{\pm.002}$ | $2.712^{\pm.007}$ | $9.592^{\pm.093}$ |
| 5.0 | 1.5 | $0.037^{\pm.002}$ | $0.563^{\pm.003}$ | $0.755^{\pm.002}$ | $0.843^{\pm.002}$ | $2.693^{\pm.008}$ | $9.496^{\pm.094}$ |

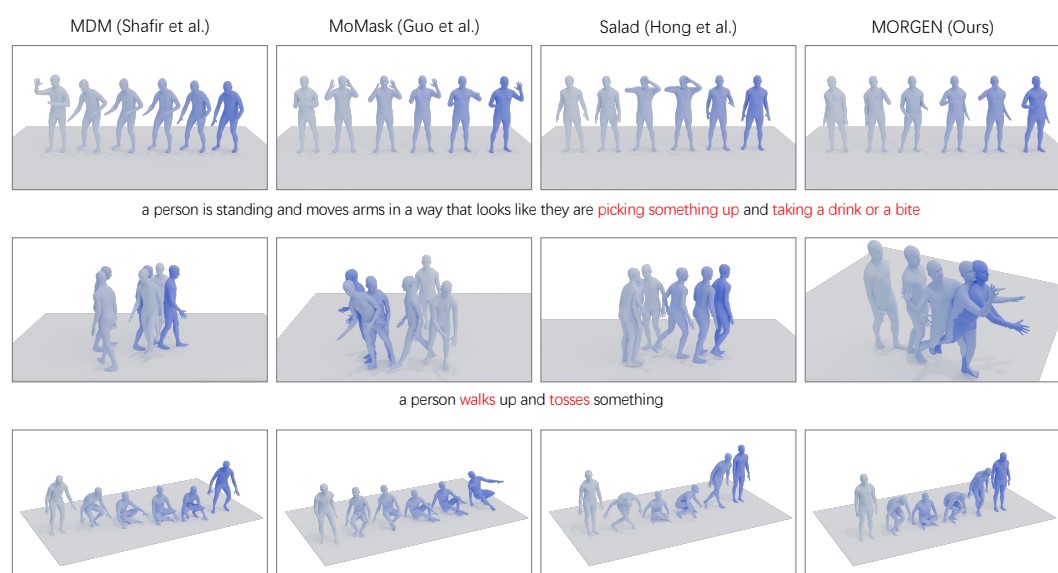

Figure 3: Qualitative evaluation on the HumanML3D Dataset. Please zoom in for details.

observed that baseline methods often fail to faithfully execute the entire set of actions described in the text. For example, in the second row ("a person walks up and tosses something"), most methods only execute the walking motion. Additionally, some outputs display distortions, such as unnatural transitions—in the third row, MoMask during sitting down and Salad during standing up. In contrast, our method successfully completes all actions described by each text prompt, demonstrating a high degree of semantic accuracy and realism.

## 5 CONCLUSION

In this work, we present **MORGEN**, a novel framework that leverages motion reconstruction to regularize text-driven motion diffusion models. Our approach focuses on learning a motion-centric latent space via motion reconstruction, specifically designed to capture essential motion dynamics while achieving high semantic resolution. This latent space serves as intermediate supervision for text-to-motion generation, bridging the representational gap between abstract language and high-dimensional, kinematically constrained human motion. We further present Reconstructive Error Guidance (REG), a technique that mitigates error accumulation during sampling by exploiting the diffusion model's inherent self-correcting ability. Experimental results show that MORGEN achieves state-of-the-art performance on standard benchmarks. In the future, we plan to extend this approach by training the motion reconstruction branch on larger, unlabeled motion datasets to obtain a more generalized motion latent space and enable the generation of more diverse motions.

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

Table 6: Effect of encoder latent dimension $d_E$.

| $d_E$ | FID↓ | R-Precision | | | MM Dist↓ | Diversity↑ |
|---|---|---|---|---|---|---|
| | | Top 1 | Top 2 | Top 3 | | |
| 128 | $0.046^{\pm.003}$ | $0.552^{\pm.003}$ | $0.743^{\pm.003}$ | $0.833^{\pm.002}$ | $2.749^{\pm.008}$ | $9.502^{\pm.073}$ |
| 192 | $0.050^{\pm.003}$ | $0.559^{\pm.002}$ | $0.748^{\pm.003}$ | $0.836^{\pm.002}$ | $2.717^{\pm.008}$ | $9.573^{\pm.054}$ |
| 256 | $0.037^{\pm.002}$ | $0.563^{\pm.003}$ | $0.755^{\pm.002}$ | $0.843^{\pm.002}$ | $2.693^{\pm.008}$ | $9.496^{\pm.094}$ |
| 320 | $0.049^{\pm.003}$ | $0.565^{\pm.002}$ | $0.760^{\pm.003}$ | $0.848^{\pm.002}$ | $2.691^{\pm.009}$ | $9.554^{\pm.079}$ |
| 512 | $0.061^{\pm.003}$ | $0.560^{\pm.003}$ | $0.753^{\pm.002}$ | $0.842^{\pm.002}$ | $2.721^{\pm.005}$ | $9.610^{\pm.084}$ |

Table 7: Effect of objective weights.

| $w_{\text{latent}}$ | $w_{\text{sr}}$ | FID↓ | R-Precision | | | MM Dist↓ | Diversity↑ |
|---|---|---|---|---|---|---|---|
| | | | Top 1 | Top 2 | Top 3 | | |
| 1.0 | 1.0 | $0.037^{\pm.002}$ | $0.563^{\pm.003}$ | $0.755^{\pm.002}$ | $0.843^{\pm.002}$ | $2.693^{\pm.008}$ | $9.496^{\pm.094}$ |
| 1.0 | 0.5 | $0.055^{\pm.003}$ | $0.560^{\pm.003}$ | $0.751^{\pm.003}$ | $0.841^{\pm.002}$ | $2.703^{\pm.009}$ | $9.482^{\pm.080}$ |
| 1.0 | 0.1 | $0.063^{\pm.004}$ | $0.555^{\pm.004}$ | $0.747^{\pm.002}$ | $0.837^{\pm.002}$ | $2.729^{\pm.006}$ | $9.532^{\pm.074}$ |
| 1.0 | 0.0 | $0.109^{\pm.005}$ | $0.533^{\pm.003}$ | $0.722^{\pm.002}$ | $0.815^{\pm.002}$ | $2.859^{\pm.009}$ | $9.508^{\pm.084}$ |
| 0.5 | 1.0 | $0.032^{\pm.002}$ | $0.561^{\pm.003}$ | $0.751^{\pm.002}$ | $0.839^{\pm.002}$ | $2.716^{\pm.007}$ | $9.487^{\pm.084}$ |
| 0.1 | 1.0 | $0.056^{\pm.002}$ | $0.534^{\pm.003}$ | $0.730^{\pm.002}$ | $0.822^{\pm.001}$ | $2.839^{\pm.007}$ | $9.465^{\pm.062}$ |
| 0.0 | 1.0 | $0.422^{\pm.016}$ | $0.476^{\pm.003}$ | $0.673^{\pm.003}$ | $0.778^{\pm.002}$ | $3.134^{\pm.010}$ | $9.451^{\pm.075}$ |

# A APPENDIX

## A.1 THE USE OF LARGE LANGUAGE MODELS

This work utilized Large Language Models (LLMs) as auxiliary tools to support our research process. Specifically, LLMs were employed for text refinement and language polishing to improve clarity and readability.

We emphasize that all LLM-generated or LLM-refined text underwent thorough review and revision by the authors to ensure accuracy, appropriateness, and alignment with our research findings.

The authors take full responsibility for all content presented in this paper and have employed LLMs rigorously and responsibly to enhance, rather than replace, human scholarly judgment and expertise.

## A.2 ANALYSIS OF ENCODER LATENT DIMENSION

The dimensionality of the encoder's latent space, $d_E$—which determines the size of the motion and text latents—is a key hyperparameter in our model. A larger $d_E$ can capture more intricate details but increases the model's parameter count and the risk of overfitting, while a smaller $d_E$ may result in information loss. In our main experiments, we set $d_E$ to 256. Here, we further explore how varying $d_E$ influences MORGEN's performance.

As shown in Table 6, altering $d_E$ leads to only minor fluctuations in performance, indicating that MORGEN is relatively robust to this hyperparameter. Interestingly, even when $d_E$ is halved to 128, MORGEN's performance only decreases slightly. This suggests that the learned latent space is highly compact.

## A.3 SENSITIVITY ANALYSIS OF OBJECTIVE WEIGHTS

We investigate the impact of the weights $w_{\text{latent}}$ and $w_{\text{sr}}$, which correspond to motion-centric latent alignment and self-regularization, respectively. The results are presented in Table 7. It can be observed that setting either weight to zero results in a significant performance drop. However, as long as both weights are nonzero, changes in their values have only a minor effect on performance. These findings highlight the importance of each objective component and demonstrate MORGEN's robustness to variations in weight assignment.

## A.4 INFERENCE EFFICIENCY

The proposed Reconstructive Error Guidance (REG) introduces an additional reconstruction step for previous predictions during inference, which increases inference time. However, experiments

Table 8: Experiments on inference efficiency.

| Method | enable REG at step $t$ | AITS↓ | FID↓ | R-Precision Top 1 | Top 2 | Top 3 |
|---|---|---|---|---|---|---|
| MDM-50steps | None | 0.490 | $0.398^{\pm.010}$ | $0.456^{\pm.002}$ | $0.646^{\pm.003}$ | $0.752^{\pm.002}$ |
| MORGEN-20steps | None | 0.226 | $0.126^{\pm.005}$ | $0.560^{\pm.003}$ | $0.751^{\pm.002}$ | $0.842^{\pm.002}$ |
| | [46,44] | 0.235 | $0.088^{\pm.003}$ | $0.559^{\pm.003}$ | $0.753^{\pm.002}$ | $0.842^{\pm.002}$ |
| | [46, 44, 41, 39] | 0.261 | $0.057^{\pm.002}$ | $0.560^{\pm.003}$ | $0.753^{\pm.002}$ | $0.843^{\pm.002}$ |
| | [46, 44, 41, 39, 36, 34] | 0.284 | $0.046^{\pm.002}$ | $0.560^{\pm.003}$ | $0.754^{\pm.002}$ | $0.843^{\pm.002}$ |
| | All except the inital step | 0.398 | $0.037^{\pm.002}$ | $0.563^{\pm.003}$ | $0.755^{\pm.002}$ | $0.843^{\pm.002}$ |

show that MORGEN requires substantially fewer inference steps than most commonly used motion diffusion models. As discussed in Section 4.2, training uses $T = 50$ diffusion steps. For efficient inference, we automatically select 20 denoising steps by linearly spacing them within the interval $[0, \ldots, T - 1]$, resulting in the following indices:

$$t = [49, 46, 44, 41, 39, 36, 34, 31, 28, 26, 23, 21, 18, 15, 13, 10, 8, 5, 3, 0]$$

We compare MDM-50steps and several configurations where REG is applied at different steps, reporting both their average inference time per sentence (AITS) (Chen et al., 2023) and the resulting generation quality. AITS is calculated on the HumanML3D test set by setting the batch size to 1 and excluding model and dataset loading time. Note that MDM-50steps is an improved variant of the original MDM, offering higher inference efficiency and better generation results compared to those reported in the original paper (Shafir et al., 2023).

The results summarized in Table 8 show that, due to fewer inference steps, MORGEN achieves significantly higher efficiency even when REG is enabled at every step (except the initial one, which lacks a previous prediction). Notably, enabling REG only in the early denoising steps already leads to a marked improvement in FID. This supports our claim in the introduction that early denoising steps—responsible for recovering motion from nearly pure noise—are particularly prone to generating error patterns, and thus benefit most from the application of REG.

