# OpenReview forum: "Reconstruction for Generation: Regularizing Motion Diffusion Models with Motion Reconstruction"
_ICLR.cc/2026/Conference — ICLR 2026 Conference Withdrawn Submission_

### Official Review · Reviewer_wXvw · 2025-10-30

**Soundness:** 2
**Presentation:** 1
**Contribution:** 1
**Rating:** 2
**Confidence:** 5

**Summary:**

This paper introduces MORGEN (MOtion Reconstruction for GENeration), a novel framework to enhance text-driven human motion generation with diffusion models. It addresses two limitations: the gap between text and motion representations and the accumulation of errors during the denoising process.

**Strengths:**

- The paper is well written.

- The research problem is important.

- Effective method and good results.

**Weaknesses:**

- My main concern lies in the limited contribution and novelty of this work, because most of the motivation and claimed contribution have been proposed by previous work. For instance, the contrastive loss (Self-Regularization claimed by authors) has been used in HumanTOMATO (ICML 2024) for enhancing the T-M alignment. Besides, the latent alignment of the motion and text has been explored by TMR (ICCV 2023). These insights have been claimed by previous studies and make the authors' contribution quite limited. Please discuss the difference with them.

- The demo results look quite poor. No mesh, no foot contact visualization.

- The R-P metric is quite higher than the ground truth. How do authors think this sounds enough for the evaluation?

- The method seems to work on the diffusion model. Have authors tested on several diffusion models? Such as MotionLCM, MLD?

**Questions:**

see above

---

### Official Review · Reviewer_589g · 2025-10-31

**Soundness:** 2
**Presentation:** 3
**Contribution:** 2
**Rating:** 4
**Confidence:** 4

**Summary:**

This paper introduces MORGEN, a new framework for text-to-motion generation that aims to solve two primary problems with existing motion diffusion models: 1) the "representational gap" from pre-trained text encoders (like CLIP) that lack motion-specific dynamic information, and 2) "error accumulation" that occurs during the iterative denoising process.

To address the first problem, MORGEN co-trains a motion reconstruction branch, creating a motion-centric latent space. This space is shaped by a "self-regularization" loss to improve its structure and is used as an intermediate target for the text embeddings via a "motion-centric latent alignment" loss.

To address the second problem, the paper proposes a new inference-time strategy called Reconstructive Error Guidance (REG). At each denoising step, REG uses the motion reconstruction branch to reconstruct the previous (and presumably more error-prone) estimate. It then calculates the residual between the current, more accurate prediction and this "error" reconstruction, and uses this residual to guide the sampling process away from past errors. The authors demonstrate state-of-the-art results, particularly in FID, on the HumanML3D and KIT-ML benchmarks.

**Strengths:**

1. Excellent Quantitative Performance: The method achieves state-of-the-art results on the HumanML3D benchmark (Table 1). The most notable result is the FID of 0.037, which not only surpasses prior diffusion-based methods but also outperforms most VQ-VAE models, which have historically been stronger on this metric. This demonstrates that a well-regularized diffusion model can achieve top-tier motion fidelity.

2. Effective Inference-Time Guidance: The proposed Reconstructive Error Guidance (REG) is shown to be highly effective. The ablation in Table 5 clearly demonstrates that REG is the primary driver for the large improvement in FID (e.g., dropping the FID from 0.126 to 0.037 when combined with CFG). While guidance mechanisms are common, this specific approach of using a reconstruction conditioned on the previous step's latent as a negative reference is a novel and empirically successful contribution.

**Weaknesses:**

1. Unsupported Motivation (Representational Gap): The paper's first motivation hinges on the claim that pre-trained text models like CLIP "lack motion-specific information" (lines 66-69). This is presented as fact, but I see no strong evidence or analysis to support it. What specific information is missing? If this gap is so critical, it would imply that text encoders for text-to-video models also need fundamental changes, yet many successful T2V models use standard text-to-image encoders. The paper needs to better substantiate this claim to justify the necessity of its motion-centric latent alignment.

2. Misleading Motivation (Error Accumulation): My second major concern is with the "error accumulation" motivation (lines 71-72). The paper cites Chung et al. (2022) to support the claim that early denoising steps are "prone to generating error patterns." This citation seems misplaced. The Chung paper's "error" is an accumulation during a back-and-forth (noise/denoise) inpainting process, which is mechanistically different from a standard, unidirectional denoising schedule. Furthermore, if this were a fundamental flaw in the denoising process, it should plague all text-to-image and text-to-video diffusion models. The authors don't provide a convincing argument for why this is a unique or critical problem for motion generation that standard models haven't already implicitly handled.

3. Unconvincing Qualitative Results: The visualized results (Figure 3) are not compelling. The text prompts are very simple (e.g., "a person walks up and tosses something"). More importantly, I can hardly tell the difference between the outputs from MORGEN and the baselines; they all seem to generate similar, simple motions. The paper notes that baselines like MDM can generate trajectory movements, but none are shown. The visualizations ultimately fail to support the paper's core claims. They do not show MORGEN alleviating "error patterns" in a visible way, nor do they demonstrate that its text encoder captures more complex "motion-specific" information than the baselines.

**Questions:**

1. Evidence for Motivation 1: Following up on Weakness #1, can the authors provide any empirical analysis (e.g., a probing task or a t-SNE visualization) that actually demonstrates this "representational gap"?

2. Complex Prompts: Can the authors provide qualitative results for more complex and challenging text prompts? For example, prompts that involve longer sequences of distinct actions, complex body-part interactions, or more nuanced adverbs that would truly test the "motion-specific" understanding of the model.

3. On the "Motion-Centric" Space: The paper argues for aligning the text space to the motion space (using $\beta=0.01$ in Eq. 6) because this "prioritizing motion space leads to stronger performance." But the motion encoder $E_m$ is also trained from scratch. What makes this learned motion latent space inherently superior to the pre-trained text space as an anchor, given the final task is text-conditioned generation?

---

### Official Review · Reviewer_2ZNj · 2025-10-31

**Soundness:** 3
**Presentation:** 3
**Contribution:** 2
**Rating:** 2
**Confidence:** 4

**Summary:**

This paper proposes MORGEN, a diffusion-based framework for text-to-motion generation that tackles two core issues: the weak motion understanding of pre-trained text encoders and error accumulation during denoising. MORGEN co-trains a motion reconstruction branch with self-regularization and motion-centric latent alignment to bridge text and motion representations. It further introduces Reconstructive Error Guidance (REG), which uses reconstructed residuals at inference to reduce accumulated noise.

**Strengths:**

- The proposed solution has a clear motion-centric design: A two-branch architecture (motion reconstruction + text-to-motion) with self-regularization (Eq. 5) to separate motion latents and motion-centric latent alignment (Eq. 6) to map text to motion space; this is optimized jointly with the diffusion decoder.

- The proposed method was evaluated on two common benchmarks and compared with several sota methods.

**Weaknesses:**

1. The paper acknowledges two-stream predecessors and joint language–motion spaces (e.g., Ahuja & Morency; TEMOS/TMR) and argues for motion-centric alignment with controlled gradient flow rather than a symmetric joint space. It would help to explicitly contrast training signals and gradient paths (e.g., where MORGEN’s L_sr and β-gated L_latent differ from prior contrastive/KL alignments) and add a side-by-side schematic or ablation versus a fully joint space baseline to solidify the novelty claim.

2. REG is presented as a heuristic without a concrete theoretical justification. The text frames REG via a self-correction hypothesis and residual amplification but stops short of a deeper analysis of score geometry or sampling trajectories. It would be beneficial to consider adding (a) trajectory visualizations of xt with/without REG across timesteps, (b) a small error decomposition (kinematic smoothness, foot-skate, jerk) across steps.

3. The implementation fixes DistilBERT for text and TEMOS-style encoders for motion. The results do not test other common text encoders (e.g., CLIP/T5) or encoder capacities, beyond latent-dim sweeps.

4. The performance gains of the proposed method over several recent state-of-the-art methods appear relatively modest, especially on HumanML3D. Some metrics are fall behind VAE based methods.  It is therefore difficult to assess how much of the advantage stems from the proposed components versus training or tuning choices. The supplementary materials include a few qualitative video demos, but comparisons to baseline methods are limited.

5. REG adds extra decoding calls per step; while the paper reports AITS and shows that using 20 steps with selective REG yields better quality than MDM-50 (Table 8), it would help to report wall-clock (sec/sample) and FLOPs for key configurations, and to clarify which steps benefit most (early vs. late) in a plot.

6. The qualitative comparison is very important for the motion generation application. However, the qualitative results (Fig. 3 only) are very limited.

7. It is suggested to include more recent SoTA methods for comparison, such as [a] and [b].

[a] Tlcontrol: Trajectory and language control for human motion synthesis

[b] MaskControl: Spatio-Temporal Control for Masked Motion Synthesis

8. It would be good to also discuss the limitations of the proposed work.

**Questions:**

Please refer to the weakness section.

---

### Official Review · Reviewer_d5t6 · 2025-11-01

**Soundness:** 4
**Presentation:** 3
**Contribution:** 3
**Rating:** 4
**Confidence:** 4

**Summary:**

This paper proposes MORGEN, which employs a two-stream pipeline (text→motion + motion-reconstruction branch) for simultaneously motion reconstruction and generation. During training, the paper uses a self-regularization loss and a motion-centric latent alignment loss along with reconstruction loss and generation loss for a more regularized motion-text latent. During inference, this paper adds a reconstructive error guidance to correct the accumulation error.

**Strengths:**

1. Aligning motion/text spaces + a reconstruction-aware inference guidance is intuitive and techniquely sound.
2. Quantitative results are SOTA-ish on the reported benchmarks, and the 20-step sampler is efficient.

**Weaknesses:**

1. During training, there is no augmentation to make Em robust to artifacts. But during inference, the Em is applied to encode the erroneous generated result at timestep t, so that the encoded motion condition z_m may be not reliable

2. For the text-motion alignment part, I suggest the author do some targeted ablation study, such as motion-text retrieval, to verify the effectiveness of the motion-text latent alignment itself. Are they truly aligned after training?

3. Which evaluator are you using? Do all methods use the same evaluator under the same representation and settings?

4. Missing strong baselines results. Compare against recent, stronger works (MARDM [1], Motion Streamer [2], MotionLCM v2 [3]) with both qualitative and quantitative results. Addionally, since FID in text-to-motion is proven to be fragile and sometimes misaligned with human judgment, include a human study, and consider add a different evaluator (e.g., the non-redundant representation evaluator proposed in MARDM [1]).

4. The videos in supplement is not solid. Why for comparison baselines you show root trajectories, but not for your own results? And skeleton-only visualization results makes the motion differences hard to see. From the provided results, visual improvements don’t look significant, especially compared with MoMask. I suggest the author to render the results on a human mesh like existing work such as MoMask [4].

Reference

[1] Rethinking Diffusion for Text-Driven Human Motion Generation

[2] MotionStreamer: Streaming Motion Generation via Diffusion-based Autoregressive Model in Causal Latent Space

[3] MotionLCM-V2: Improved Compression Rate for Multi-Latent-Token Diffusion

[4] MoMask: Generative Masked Modeling of 3D Human Motions

**Questions:**

Refer to weaknesses.

---

### Note · Authors · 2025-11-20

I have read and agree with the venue's withdrawal policy on behalf of myself and my co-authors.